# Monocyte-derived transcriptome signature indicates antibody-dependent cellular phagocytosis as a potential mechanism of vaccine-induced protection against HIV-1

Shida Shangguan[1,2], Philip K Ehrenberg[1], Aviva Geretz[1,2], Lauren Yum[1,2], Gautam Kundu[1,2], Kelly May[1,2], Slim Fourati[3], Krystelle Nganou-Makamdop[4], LaTonya D Williams[5], Sheetal Sawant[5], Eric Lewitus[1,2], Punnee Pitisuttithum[6], Sorachai Nitayaphan[7], Suwat Chariyalertsak[8], Supachai Rerks-Ngarm[9], Morgane Rolland[2], Daniel C Douek[4], Peter Gilbert[10], Georgia D Tomaras[5], Nelson L Michael[1], Sandhya Vasan[1,2], Rasmi Thomas[1]*

[1]US Military HIV Research Program (MHRP), Walter Reed Army Institute of Research, Silver Spring, United States; [2]Henry M. Jackson Foundation for the Advancement of Military Medicine, Bethesda, United States; [3]Department of Pathology and Laboratory Medicine, Emory University, Atlanta, United States; [4]Vaccine Research Center, NIH, Bethesda, United States; [5]Departments of Surgery, Immunology and Molecular Genetics and Microbiology, Duke University School of Medicine, Durham, United States; [6]Vaccine Trial Centre, Faculty of Tropical Medicine, Mahidol University, Bangkok, Thailand; [7]Armed Forces Research Institute of Medical Sciences, Bangkok, Thailand; [8]Research Institute for Health Sciences and Faculty of Public Health, Chiang Mai University, Chiang Mai, Thailand; [9]Department of Disease Control, Ministry of Public Health, Nonthaburi, Thailand; [10]Fred Hutchinson Cancer Research Center, Seattle, United States

*For correspondence:
rthomas@hivresearch.org

Competing interest: The authors declare that no competing interests exist.

**Abstract** A gene signature was previously found to be correlated with mosaic adenovirus 26 vaccine protection in simian immunodeficiency virus and simian-human immunodeficiency virus challenge models in non-human primates. In this report, we investigated the presence of this signature as a correlate of reduced risk in human clinical trials and potential mechanisms of protection. The absence of this gene signature in the DNA/rAd5 human vaccine trial, which did not show efficacy, strengthens our hypothesis that this signature is only enriched in studies that demonstrated protection. This gene signature was enriched in the partially effective RV144 human trial that administered the ALVAC/protein vaccine, and we find that the signature associates with both decreased risk of HIV-1 acquisition and increased vaccine efficacy (VE). Total RNA-seq in a clinical trial that used the same vaccine regimen as the RV144 HIV vaccine implicated antibody-dependent cellular phagocytosis (ADCP) as a potential mechanism of vaccine protection. CITE-seq profiling of 53 surface markers and transcriptomes of 53,777 single cells from the same trial showed that genes in this signature were primarily expressed in cells belonging to the myeloid lineage, including monocytes, which are major effector cells for ADCP. The consistent association of this transcriptome signature with VE represents a tool both to identify potential mechanisms, as with ADCP here, and to screen novel approaches to accelerate the development of new vaccine candidates.

## Introduction

The only HIV vaccine in ongoing human efficacy trials employs an adenovirus serotype 26 (Ad26)-based vector, but tests a different route of infection and geographic population compared to African women in whom it failed to show efficacy (*Mega, 2019*; *NIH, 2017*; *NIH, 2021*). Vaccines from these trials were previously tested in non-human primates (NHPs) and showed partial protection from infection (*Barouch et al., 2015*; *Barouch et al., 2018*). To date, the pivotal RV144 phase three human efficacy trial conducted in Thailand is the only vaccine to show any protection against HIV (*Rerks-Ngarm et al., 2009*). This vaccine used a canary-pox ALVAC-based vector with a bivalent gp120 protein boost. Although neither of the vaccine regimens using Ad26 or canary-pox viral vectors were fully efficacious, there is some consensus that current preventive and treatment methods along with a moderately effective vaccine could potentially reduce the HIV pandemic (*Anderson et al., 1995*; *Andersson et al., 2007*; *Fauci, 2017*; *Medlock et al., 2017*). While a number of correlates of vaccine protection have been described for these studies, protection mediated by the humoral immune systems, including HIV-1 specific IgG antibody titers, antibody Fc polyfunctionality, antibody interactions with HLA class II gene products, and antibody effector functions, has been key features of these partially effective vaccines (*Barouch et al., 2015*; *Barouch et al., 2018*; *Haynes et al., 2012*; *Prentice et al., 2015*).

We previously showed that a vaccine-induced gene signature identified in B cells by an unbiased transcriptome-wide RNA-seq approach associated with decreased risk against simian immunodeficiency virus (SIV)/simian-human immunodeficiency virus (SHIV) infection in NHP studies evaluating the Ad26 vaccine (*Ehrenberg et al., 2019*). This geneset was also enriched in NHP and the human RV144 trial that employed a vaccine containing the ALVAC viral vector (*Ehrenberg et al., 2019*). This gene signature is not merely a general response elicited by vaccination, as it was not enriched in the Ad26-MVA arm of the SHIV challenge in NHP that showed some protection previously (*Barouch et al., 2018*). This gene signature was initially defined when comparing differentially expressed genes (DEGs) between B cells and monocytes from vaccinated individuals in an Influenza immunogenicity trial (*Nakaya et al., 2011*). The geneset that was submitted to the molecular signature database (MSigdb) comprised the top 200 genes that were upregulated in monocytes compared to B cells. In our previous study, specific genes in this geneset were upregulated in the uninfected compared to the infected group in multiple SIV/HIV trials (*Ehrenberg et al., 2019*). Genes that were previously correlated with immunogenicity in human vaccine trials of influenza and yellow fever, including *TNFSF13 (APRIL)*, were also enriched in uninfected rhesus monkeys in the NHP studies (*Li et al., 2014*; *Nakaya et al., 2011*). Although we first identified this geneset in sorted B cells (*Ehrenberg et al., 2019*), we were able to identify the same signature associating with reduced infection in published microarray data sets from both bulk unstimulated and in vitro antigen-stimulated PBMC from three independent preclinical and clinical studies (*Fourati et al., 2019*; *Vaccari et al., 2018*; *Vaccari et al., 2016*). To determine if specific immune responses might be driving protection in conjunction with the gene signature, we examined whether this geneset associated with these responses measured in the NHP studies. We observed that this gene signature was also enriched in animals with the increased magnitude of ADCP (*Ehrenberg et al., 2019*). We propose that this gene signature is a correlate of reduced risk of infection in efficacy studies and that further investigation of the enriched genes in the geneset could potentially help uncover the mechanism of vaccine protection. Here, we investigate this gene signature further as a proxy of vaccine-induced protection in human clinical trials to identify the cellular origin, as well as to investigate potential mechanisms for the decreased risk of infection.

## Results

### Gene signature is absent in a human HIV vaccine trial that did not show efficacy

Since the gene signature associated with protection within the vaccinated group in multiple studies from different sources and regimens, we wanted to further confirm that this signature was truly associated with protection by looking for its presence or absence in a human vaccine trial that failed to show efficacy (*Supplementary file 1a*). We screened for this gene signature in whole-transcriptome data within participants vaccinated with the DNA/rAd5 HIV-1 preventive vaccine in the HVTN 505 human efficacy trial. Immunizations in this trial were halted prior to reaching the clinical endpoint due

**Table 1.** Gene signature associates with vaccine protection in multiple trials.

| Study | Vaccine regimen | Species | Partial protection | N | Method | Protective signature |
|---|---|---|---|---|---|---|
| 09–11 | Ad26/gp140 | NHP | Y | 10 | RNA-seq | Y |
| 13–19 | Ad26/gp140 | NHP | Y | 11 | RNA-seq | Y |
| 13–19 | A26/Ad26+ gp140 | NHP | Y | 12 | RNA-seq | Y |
| 13–19 | Ad26/MVA+ gp140 | NHP | Y | 9 | RNA-seq | N |
| | ALVAC-SIV/gp120 | NHP | Y | 27 | Microarray | Y |
| | DNA-SIV/ALVAC+ gp120 | NHP | Y | 12 | Microarray | Y |
| RV144 | ALVAC/gp120 | Human | Y | 170 | Microarray | Y |
| HVTN 505 | DNA/rAd5 | Human | N | 42 | RNA-seq | N |

to lack of efficacy (*Hammer et al., 2013*). When comparing infection status within vaccinated individuals, enrichment of this gene signature, as defined by the normalized enrichment score (NES), was not significant in transcriptome data from sorted B cells or monocytes 1 month after the final immunization (NES=−1.18, p=0.09 and NES=1.12, p=0.18, respectively). This finding further supports our hypothesis that this gene signature is associated with protection, as summarized in *Table 1*.

## Genes from this signature are the strongest correlate of protection in RV144

In previous analyses of NHP preclinical studies, we utilized a composite gene expression score (GES) consisting of an average of standardized expression of the specific number of enriched genes in one study to predict infection status in an independent study using the overlapping expressed genes from the first study (*Ehrenberg et al., 2019*). While this method is successful in evaluating gene signatures in studies using similar vaccine strategies, we wanted to explore this approach across different studies using diverse platforms. For each independent study, we computed a GES derived from genes within the geneset that were enriched in uninfected donors by averaging standardized expression and showed that it associates with decreased HIV-1 infection (*Figure 1*). The magnitude of the GES and total number of enriched genes present in the gene signature are specific to each study and are higher in the uninfected compared to infected animals in the two NHP preclinical trials evaluating the mosaic Ad26 vaccine (09–11 and 13–19), including the different arms of the 13–19 study (13–19 a-b) (*Figure 1A–C*). Further, in the RV144 trial, the GES of 63 enriched genes in the gene signature also was higher in the vaccinated individuals that remained uninfected (*Figure 1D*). The number of enriched genes in each study might vary due to global differences in the vaccine strategies, but we consistently observed that higher GES associated with protection from HIV acquisition. We took advantage of the composite GES measurement to compare it with the other known primary correlates of HIV-1 infection risk in the human RV144 trial. IgG antibodies binding to the variable regions 1–2 (V1V2) of the HIV-1 Envelope (Env) have been shown to correlate with decreased risk of infection, while IgA binding to Env associated with increased risk of infection (*Haynes et al., 2012*). We show that the association of the GES in RV144 is a stronger correlate of reduced risk of infection than the previously described V1V2-specific IgG antibodies (*Figure 2A*). Cumulative incidence curves of HIV-1 infection showed decreased rates of infection among vaccine recipients with high GES (*Figure 2B*). Estimated vaccine efficacy (VE) was higher among vaccine recipients with higher GES (*Figure 2C*). The distribution of area under the receiver operating characteristic curve (AUC) and accuracy suggested that GES was also able to predict HIV-1 infection (*Figure 2D*). The effect of GES was also tested in RV144 vaccine and placebo participants who became infected during the trial (*Rolland et al., 2012*). If the GES was associated with VE, we would expect that vaccinees with a high GES would not get infected, hence vaccinees who became infected should have lower GES than placebo participants (who reflect the entire distribution of GES). This was observed across 43 breakthrough participants, with a significant difference among participants infected with single HIV-1 founder variants (N=29) (*Figure 2—figure supplement 1*). These findings strengthen the hypothesis that the GES is associated with VE.

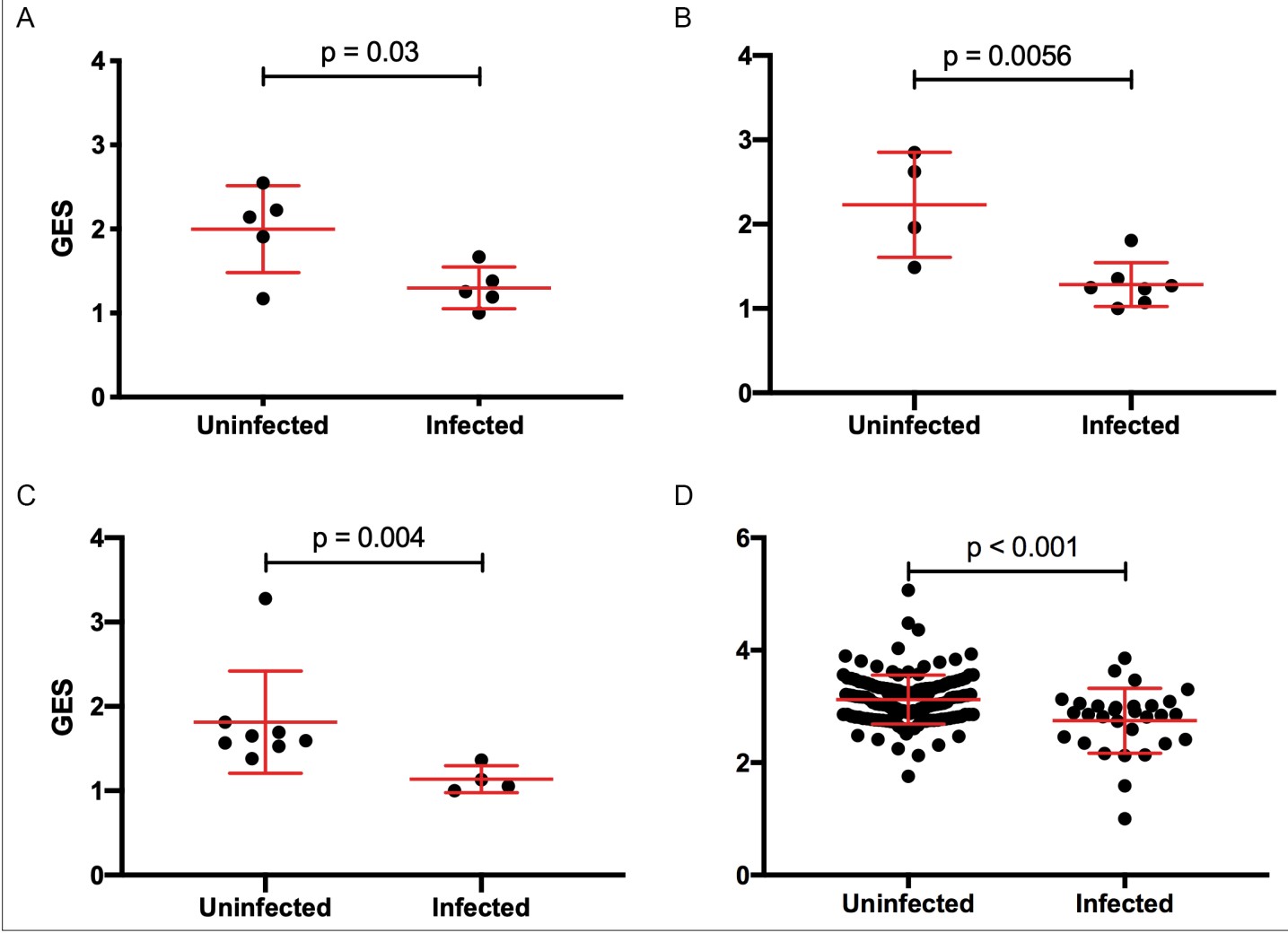

**Figure 1.** Composite gene expression scores (GES) are higher in the uninfected compared to infected groups. GES computed from enriched genes in the geneset is higher in the uninfected compared to infected vaccinated NHP and humans. (**A**) Ad26/gp140 (09–11 NHP SIV challenge study, 58 enriched genes, N=10), (**B**) Ad26/gp140 (13–19 NHP SHIV challenge study, 58 enriched genes, N=11), (**C**) Ad26/Ad26+ gp140 (13–19 NHP SHIV challenge study, 68 enriched genes, N=12), and (**D**) ALVAC/gp 120 (RV144 human efficacy trial, 63 enriched genes, N=170). Statistical significance was calculated by either Mann-Whitney or unpaired t-test. NHP, non-human primate; SHIV, simian-human immunodeficiency virus; SIV, simian immunodeficiency virus.

## Gene signature associates with an antibody effector function in a human vaccine trial

Immune responses correlating with this signature can provide additional insights into mechanisms that could be harnessed to improve vaccine design. We previously showed in the NHP studies that the protective gene signature that was enriched in uninfected monkeys after Ad26/gp140 vaccination was also associated with higher magnitude of ADCP (*Ehrenberg et al., 2019*). In the RV144 human trial a number of immunological parameters were previously measured as part of the immune-correlates analysis, but not ADCP. The RV306 immunogenicity trial that employed a similar prime-boost RV144 vaccine regimen with additional late boosts provided us with a unique opportunity to test if the gene signature was associated with ADCP (*Pitisuttithum et al., 2020*). We generated transcriptome-wide gene expression data from peripheral blood 2 weeks after the RV144 vaccine regimen (prior to the additional boosts) and assessed for enrichment of the gene signature with the magnitude of ADCP measured at the same time point in 24 participants. The gene signature with 118 enriched genes was significantly associated with higher magnitude of ADCP (NES=3.0, p<0.001) (*Figure 3A*, *Supplementary file 1b*). Using the same geneset, 93 genes were found to be enriched in a subset of overlapping

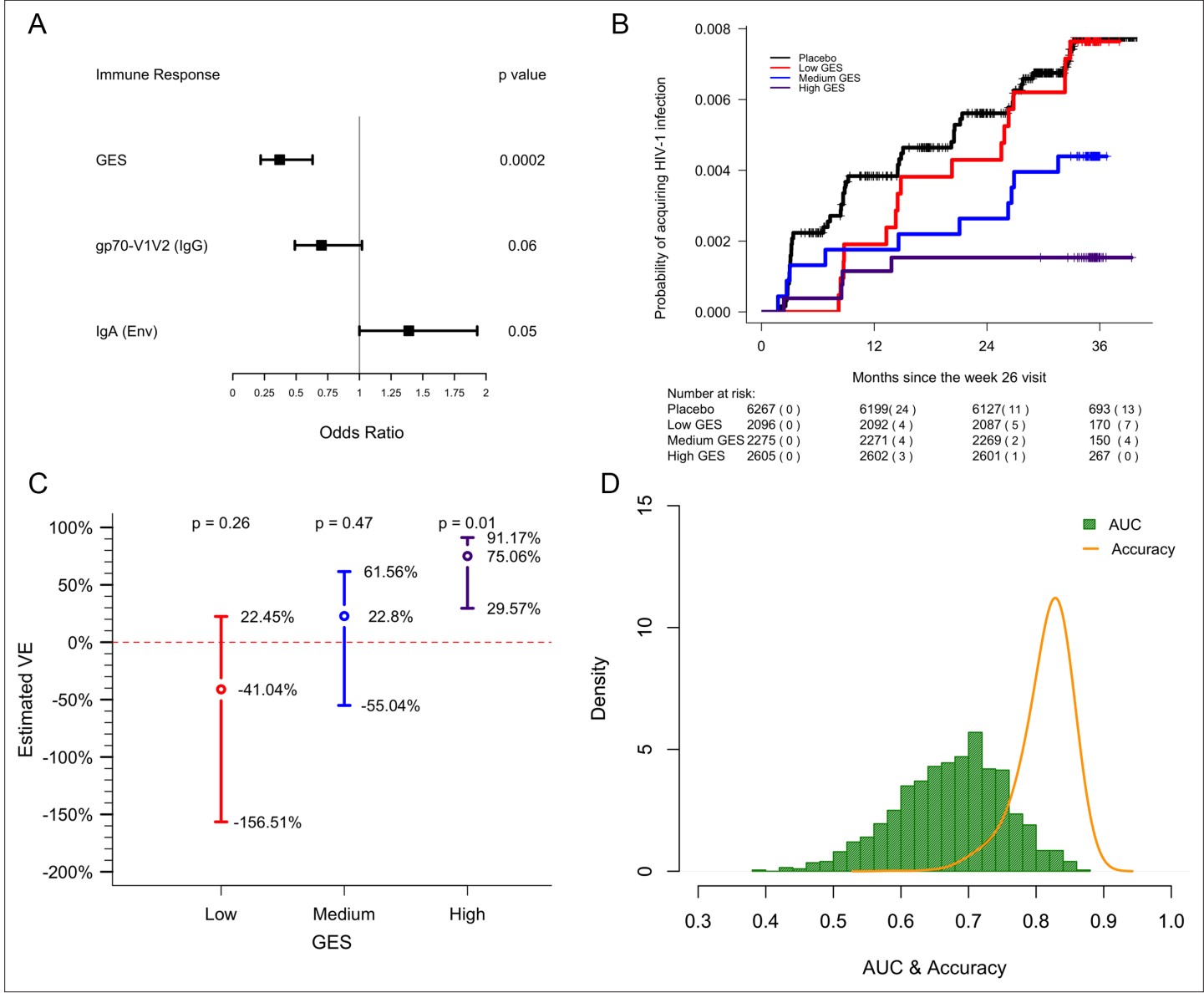

**Figure 2.** GES is a stronger correlate of reduced risk of infection in RV144. A GES of the 63 enriched genes in the RV144 study was examined as a continuous variable (N=170). (**A**) GES is associated with lower odds of HIV acquisition compared to the other two primary correlates of risk. Variables were measured at week 26, 2 weeks post last vaccination. For each variable, the OR is reported per 1-SD increase. Transcriptome data was available only in a subset of the 246 donors. (**B**) Probability of acquiring HIV-1 is lower in individuals with higher GES. (**C**) Vaccine efficacy is increased significantly in individuals with high GES. (**D**) Distribution of AUC and accuracy plotted after repeating the process 1000 times showed that GES could predict HIV-1 infection with AUC of 0.67±0.08 and with accuracy of 0.81±0.04. GES, gene expression score.

The online version of this article includes the following figure supplement(s) for figure 2:

**Figure supplement 1.** Association of the GES with HIV-1 breakthrough infections in a human vaccine trial.

participants (N=21), where samples were collected 3 days after the RV144 immunizations (NES=2.5, p<0.001) (*Figure 3A*, *Supplementary file 1b*). The model built using ADCP GES from day 3 was able to predict ADCP responses measured 2 weeks after the last vaccination with an accuracy of 0.71. The receiver operator characteristic (ROC) curve illustrates the discriminating ability of the classifier from the day 3 training data set (AUC=0.8, 95% confidence interval [CI]: 0.6–0.99, p=0.01) and the week 2 testing data set (AUC=0.73, 95% CI: 0.5–0.95, p=0.03) to predict ADCP responses (*Figure 3B*). To evaluate these findings in the context of an efficacy trial, a GES from the list of enriched genes associating with ADCP from both time points was computed in the RV144 study. ADCP GES from both

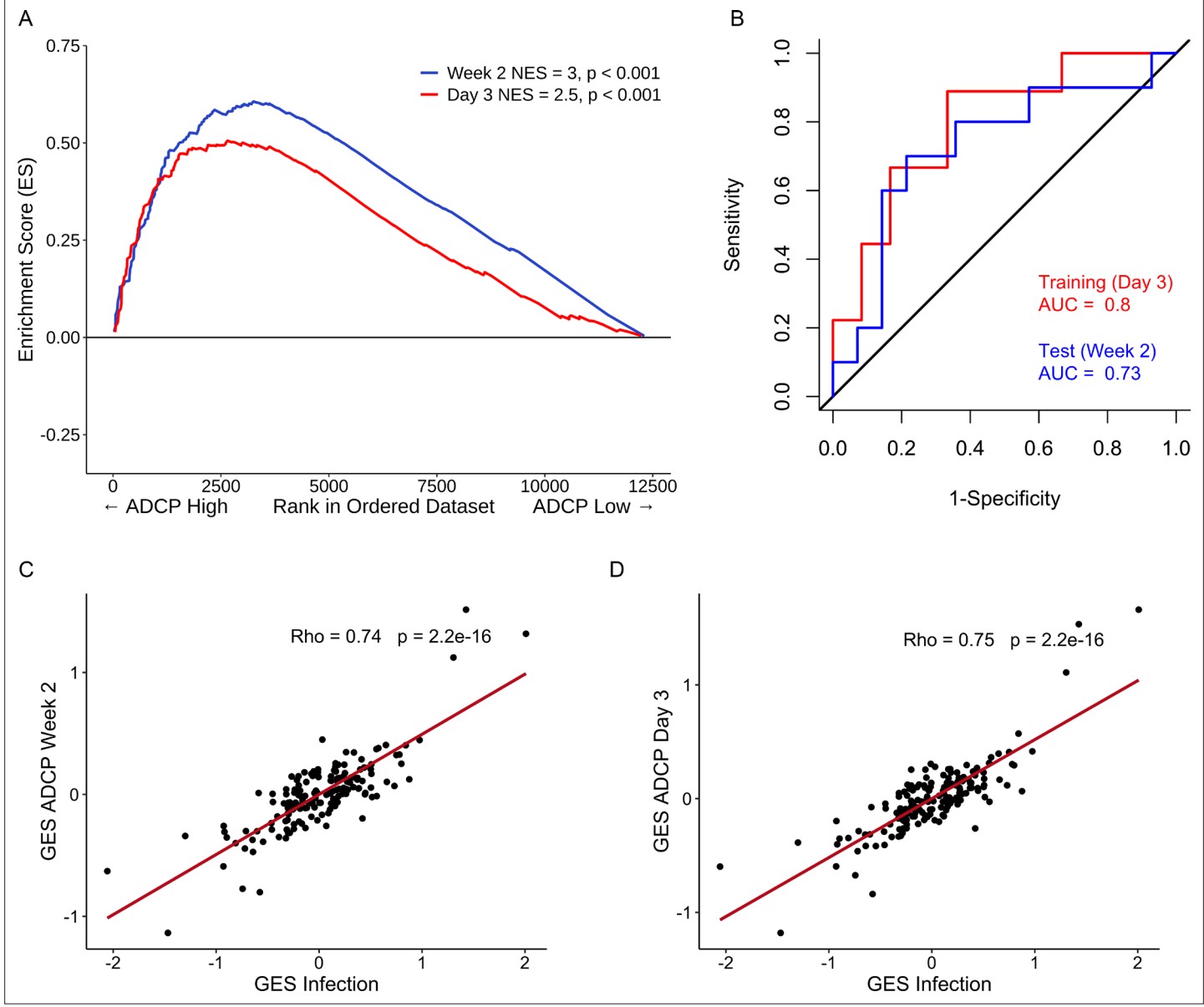

**Figure 3.** Strong relationship between functional ADCP responses in a human vaccine trial and the protective RV144 signature. The geneset that associated with protection in an efficacy study was also enriched with higher magnitude of ADCP measured 2 weeks after vaccination in an immunogenicity trial that employed the RV144 vaccine regimen. NES from RNA-seq data at time points (**A**) 2 weeks (118 enriched genes) (N=24) and 3 days (93 enriched genes) (N=21) post the RV144 vaccine regimen in the RV306 trial are indicated. (**B**) The model built using ADCP GES from day 3 was able to predict ADCP responses measured 2 weeks after the last vaccination with an accuracy of 0.71. The ROC curve illustrates the discriminating ability of the classifier from the day 3 training data set (AUC=0.8, 95% CI: 0.6–0.99, p=0.01) and the week 2 testing data set (AUC=0.73, 95% CI: 0.5–0.95, p=0.03) to predict ADCP responses. (**C**) GES computed from the enriched genes associating with ADCP correlated strongly with the protective GES in the RV144 study (N=170) at time points 2 weeks (115 enriched genes) and (**D**) 3 days (91 enriched genes) post the RV144 vaccine regimen. ADCP, antibody-dependent cellular phagocytosis; CI, confidence interval; GES, gene expression score; NES, normalized enrichment score.

time points correlated strongly with the protective RV144 GES (Rho=0.74, p=2.2e−16, Rho=0.75, p=2.2e−16) (*Figure 3C–D*). Given the strong correlation in RV144 for the enriched genes from both time points with infection status, we investigated the overlapping 82 genes from day 3 and week 2 time points in a prediction analysis (*Figure 4A–B*). In addition to being able to successfully predict ADCP magnitude, the genes also show a very clear distinction between the high versus low ADCP groups at both time points (*Figure 4C*, *Supplementary file 1b*). To gain understanding of the potential role of the 82 genes, we used GeneMANIA to explore the gene function of the 82 overlapping

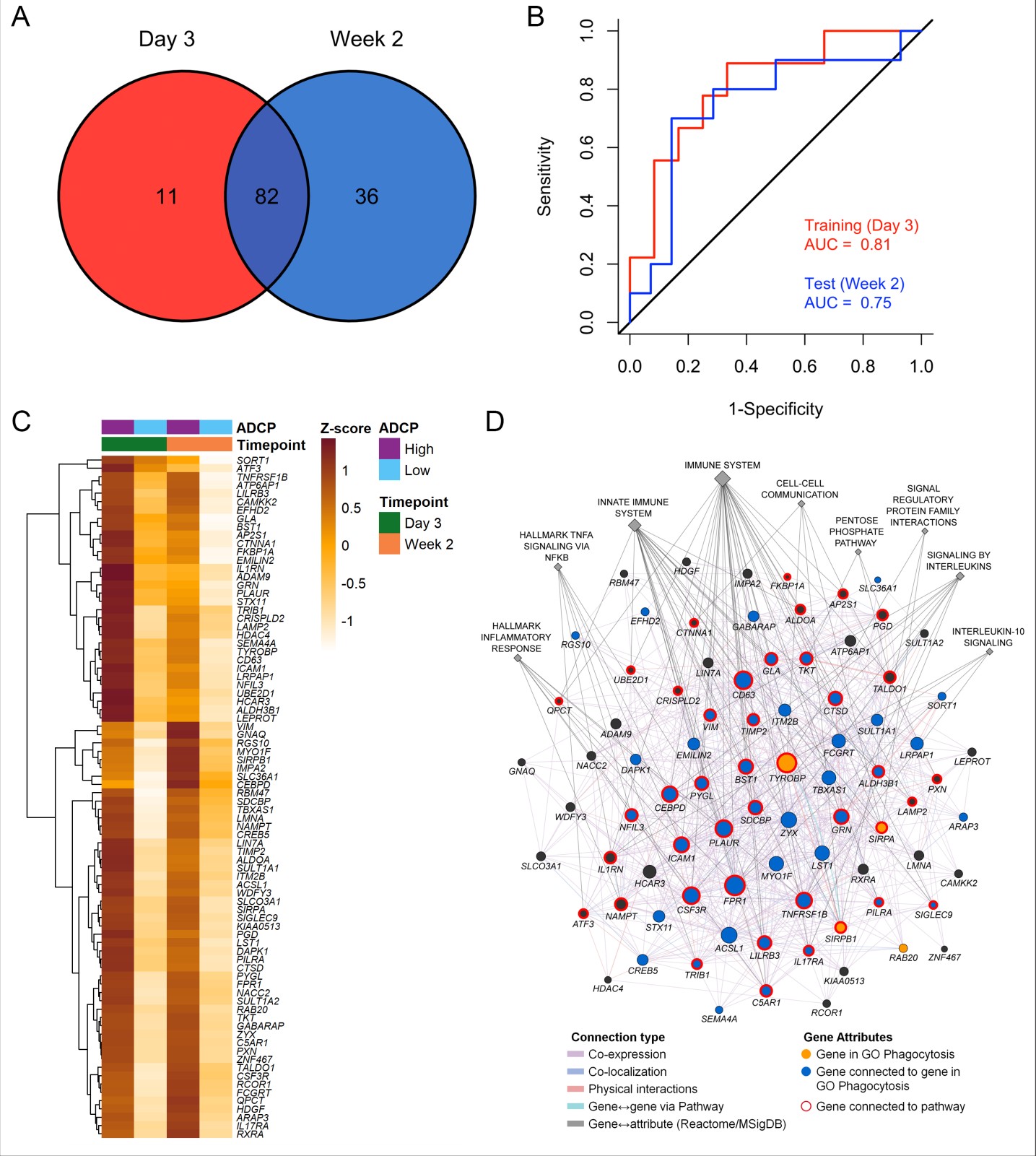

**Figure 4.** Overlapping enriched genes associating with ADCP responses. (**A**) There were 82 overlapping genes between the day 3 (N=21) and week 2 (N=24) ADCP enriched genes in the RV306 study. (**B**) The model using GES obtained from the 82 genes was also able to predict ADCP responses measured 2 weeks after vaccination with an accuracy of 0.71. The ROC curve illustrates the discriminating ability of the classifier from the day 3 training data set (AUC=0.81, 95% CI: 0.62–1, p=0.007) and the week 2 testing data set (AUC=0.75, 95% CI: 0.53–0.97, p=0.02) to predict ADCP responses. (**C**)

*Figure 4 continued on next page*

*Figure 4 continued*

Heatmap showing the hierarchical clustering of gene expression of the 82 genes (day 3 and week 2 after 4th vaccination) when stratified by magnitude of ADCP responses measured 2 weeks after the 4th immunization. (**D**) The list of 82 ADCP enriched genes was uploaded in GeneMANIA. Edges represent physical interactions, co-expression, co-localization, and shared pathways. Circles depict the 82 genes, gold circles are the four genes that belong to the gene ontology Phagocytosis pathway, blue circles are genes that are directly connected to them, diamonds indicate related pathways, and the color of the edge indicates the type of connection. ADCP, antibody-dependent cellular phagocytosis; CI, confidence interval; GES, gene expression score; ROC, receiver operator characteristic.

genes based on physical interaction, co-expression, co-localization, and shared pathways. There were 41 genes that belonged to specific top pathways including the immune system, innate immune system, signaling by interleukins, hallmark inflammatory response, hallmark TNFA signaling via NFKB, cell-cell communication, interleukin-10 signaling, signal regulatory protein family interactions, and pentose phosphate pathway (*Figure 4D*, *Supplementary file 1c*). A focused search for gene ontology (GO) terms identified four genes with phagocytosis pathway membership (*TYROBP*, *SIRPA*, *SIRPB1*, and *RAB20*) (*Supplementary file 1d*).

## Pathways and genes shared between ADCP and vaccine protection phenotypes

These findings demonstrate a strong link of the geneset with both vaccine protection and ADCP in NHP and human studies. We sought to broaden our understanding of the relationship between the different enriched genes in the geneset and establish some of the top pathways with gene membership from the different studies. Networks and associated pathways from genes that were significantly enriched with either the ADCP or infection phenotypes from the 09–11, 13–19, RV144, and RV306 studies were determined using GeneMANIA. The top pathways were the immune system, innate immune system, H1F1 TF pathway, hypoxia, TNFA signaling via NFKB, cytokine signaling, inflammation response, signaling by interleukins, and IL-10 signaling (*Figure 5A*). The genes with the most connections were *TYROBP*, *FPR1*, *CD14*, *CCR1*, *TNFRSF1B*, *CD68*, *CD63*, *CEBPD*, and *LST1*. Clustering on the enriched genes to identify highly interconnected regions in the GeneMANIA network showed that *TYROBP*, *FPR1*, *CD14*, *TNFRSF1B*, *CD68*, and *LST1* were all members of the cluster with the greatest number of genes (*Figure 5B*). There were no specific enriched genes that were common to all studies (*Supplementary file 1b*). Pathway enrichment analysis of the 63 genes in the RV144 signature revealed that the top non-redundant enriched clusters with gene membership were myeloid leukocyte activation, lysosome, and cellular response to oxidative stress genes (*Figure 5C*).

## Cellular origin of the protective genes by single-cell transcriptomics

To dissect the cellular origin of these genes, we performed simultaneous detection of mRNA and cell surface expression from single cells using the cellular indexing of transcriptomes and epitopes by sequencing (CITE-seq) technology in a subset of the vaccinated RV306 participants (*Figure 6A*). This technology allows simultaneous detection of cell surface markers and mRNA gene expression from the same single cells. Our analyses revealed that a majority of the genes in the RV144 signature were expressed in cells of the myeloid lineage, with monocyte subsets having the highest average gene expression (*Figure 6B*). A subset of 32 genes were also significantly associated with decreased risk of acquisition in a univariate analysis (odds ratio [OR ]<1.0, p<0.05, q<0.1) (*Figure 6C*). A GES of the 32 significant genes is also associated with decreased risk of acquisition, increased VE, and was able to predict infection status in RV144 (*Figure 6—figure supplement 1A-D*). A stepwise logistic regression analysis identified specific genes (*SEMA4A*, *SLC36A1*, *SERINC5*, *IL17RA*, *CTSD*, *CD68*, and *GAA*) to have independent associations with reduced risk of acquisition and was mainly expressed in the monocyte compartment (*Figure 6D*). CD14+ monocytes also had the greatest number of DEGs that were associated with ADCP, which was not dependent on the frequency of the cell subset (*Figure 6E*, *Figure 6—figure supplement 2*).

## Discussion

Though an effective vaccine has been a challenge for the HIV field, we see a glimpse of optimism in partially protective NHP and human studies (*Barouch et al., 2015*; *Barouch et al., 2013*; *Barouch*

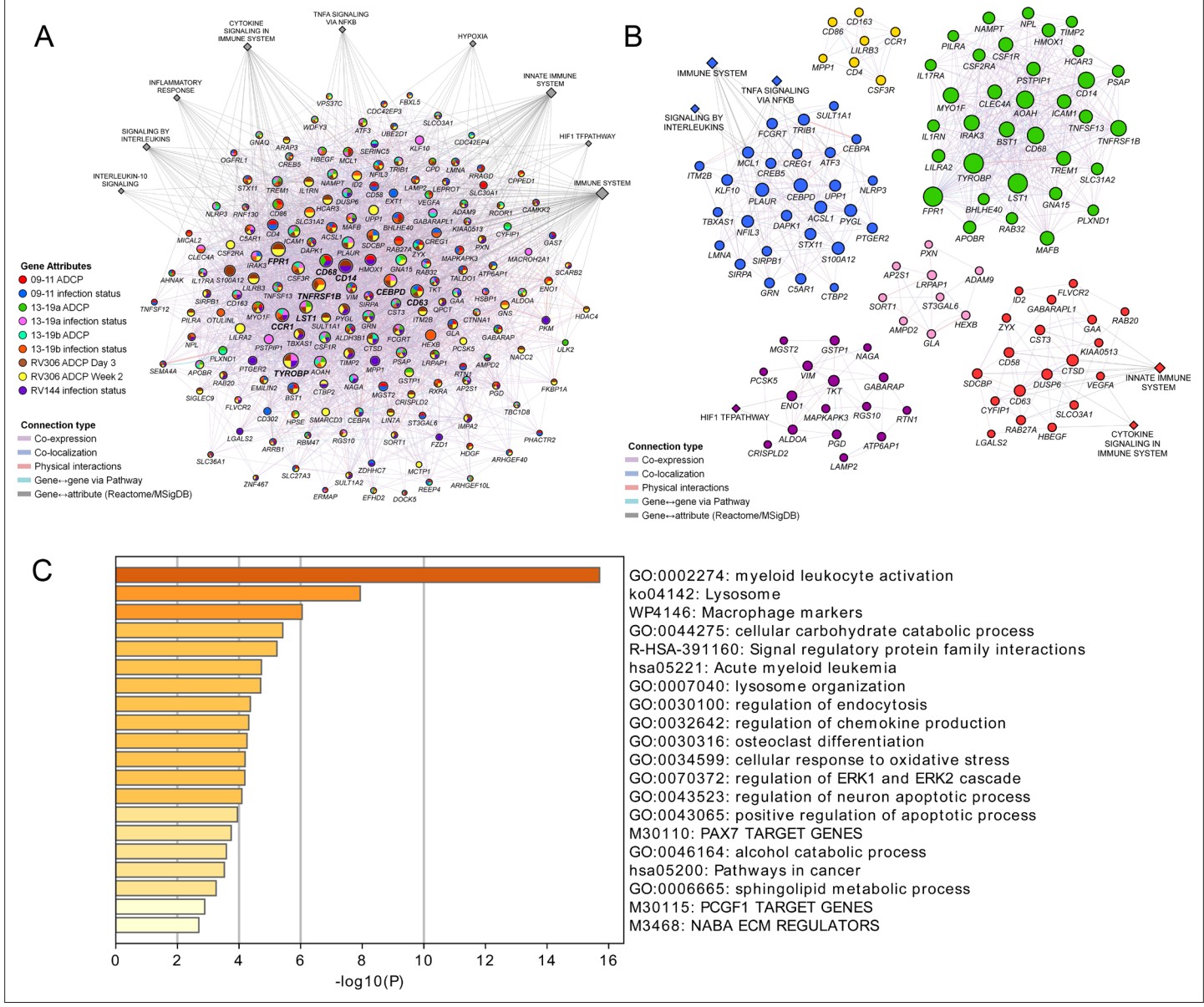

**Figure 5.** Pathway analyses of the enriched genes in the different vaccine studies. A meta-analysis of pathways including enriched genes with reduced infection or higher ADCP was performed. (**A**) Genes that were enriched in at least one of the nine ADCP or infection status analyses (178) were used as input for GeneMANIA in Cytoscape. The connections between the different genes and the top MSigDB and Reactome pathways are shown. Each gene is represented by a circle and size is proportional to the number of connections with other genes or pathways. The color of each node indicates the enrichment status in the different studies. (**B**) Clustering of the enriched genes from the different studies. The color of each node represents the membership in a cluster and size is proportional to the number of connections with other genes or pathways. (**C**) Pathway enrichment analysis results of the 63 enriched genes that associated with reduced infection in the RV144 study. ADCP, antibody-dependent cellular phagocytosis.

et al., 2018; Rerks-Ngarm et al., 2009; Vaccari et al., 2018; Vaccari et al., 2016). These studies provide a unique opportunity to identify correlates of reduced risk that could help inform protective signals and enable the design of enhanced vaccine strategies. Targeted and unbiased approaches have implicated non-neutralizing antibodies as the major correlate of reduced risk of HIV infection (Barouch et al., 2015; Barouch et al., 2013; Barouch et al., 2018; Haynes et al., 2012; Vaccari et al., 2018; Vaccari et al., 2016). We previously showed that a transcriptomic signature first identified in sorted B cells at time points prior to challenge was a correlate of protection in two NHP studies after administration of the Ad26/gp140 vaccine. This signature is also associated with the increased magnitude of ADCP in the vaccinated monkeys (Ehrenberg et al., 2019). Additionally, we identified

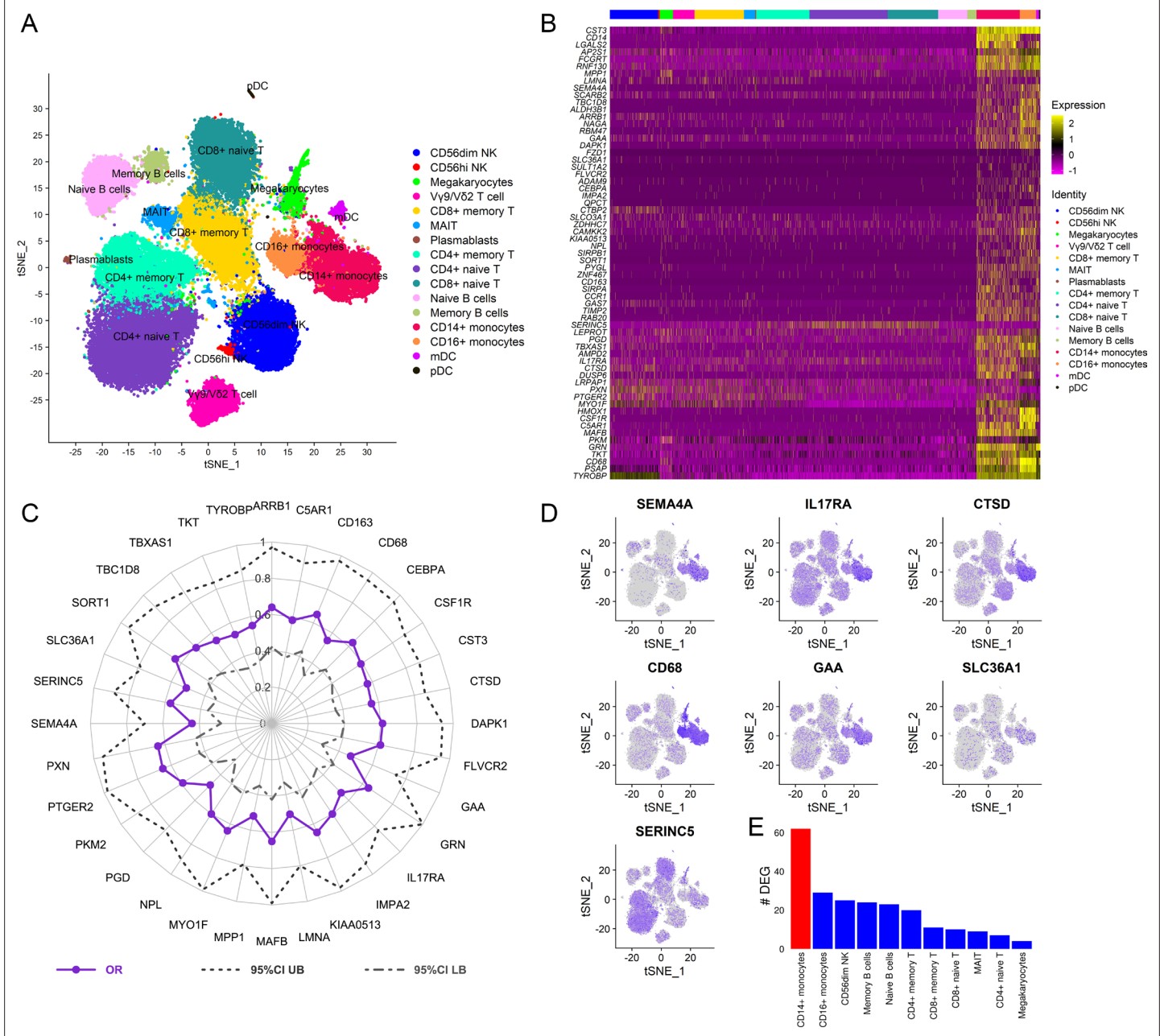

**Figure 6.** Cellular origin of the RV144 signature. Single-cell CITE-seq in vaccinated participants (N=12) who received the RV144 vaccine regimen (day 3 after last vaccination) identified expression of the genes in the signature in cells from the myeloid lineage. (**A**) Clustering based on cell surface expression of CITE-seq data. (**B**) Heatmap of the mRNA expression of the 63 genes from the RV144 signature from single cells. Columns represent single cells from different protein cell subsets and rows the mRNA gene expression. (**C**) Radar plot showing significant genes in the signature that associated with decreased risk of infection in RV144 (p<0.05, q<0.1) (N=170). (**D**) Feature plots of the expression of the most protective genes show that *SEMA4A, IL17RA, CTSD, CD68*, and *GAA* were mainly expressed in monocytes. (**E**) CD14+ monocytes had the highest number of differentially expressed genes (DEGs) when comparing high versus low ADCP (2 weeks after vaccination) from single-cell CITE-seq vaccinated participants who received the RV144 vaccine regimen (day 3 after last vaccination). ADCP, antibody-dependent cellular phagocytosis.

The online version of this article includes the following figure supplement(s) for figure 6:

**Figure supplement 1.** GES of the most significant genes is a correlate of reduced risk of infection in RV144.

**Figure supplement 2.** Frequencies of cell subsets do not differ between ADCP high and low samples.

this signature in bulk PBMC from other studies that used the ALVAC/protein regimen, suggesting that the gene signature might be an indicator of effective vaccination. In this report, we further investigated this gene signature to answer the following questions, including: (1) can the gene signature's association with protection be substantiated in additional human efficacy trials, (2) does it associate with ADCP in human trials, and (3) what is the cellular origin of the signature at the single-cell level?

The gene signature was previously associated with HIV vaccine protection in a number of studies with partial protection (*Ehrenberg et al., 2019*). We hypothesized that if this geneset was a true marker of HIV vaccine protection, it would not be enriched in a failed vaccine trial. HVTN 505 is a DNA based vaccine which, despite not showing overall efficacy in a Phase 2b trial, demonstrated both cellular and antibody effector mediated protection in specific subgroups of individuals in follow-up studies (*Fong et al., 2018*; *Janes et al., 2017*; *Neidich et al., 2019*). We performed transcriptomics on sorted cell subsets from HVTN 505 vaccinated individuals and did not observe enrichment of the gene signature, further strengthening our notion that the geneset could be a proxy for vaccine protection. Given that this signature is derived by comparing infected versus uninfected vaccinated groups, this analysis was only tested in studies with partial protection, and not in vaccine regimens where all recipients were infected.

Next, we developed a method to assess this gene signature compared to other correlates of risk in the human RV144 study. This method employs an analytical method using a GES which is computational score generated from the average expression of all genes enriched in the signature and associating with a phenotype. This method was tested across different NHP studies and RV144 and showed consistent association with reduced risk of infection based on the study-specific GES. The composite GES computed from RV144 consisting of the standardized expression of 63 genes had the strongest association with decreased risk of infection and increased vaccine efficacy. The RV144 GES was also able to accurately predict infection status in the study. This study shows that the GES composite score provides a robust analytical measurement to explore the effect of genes as a continuous variable in immune-correlates analyses, and that it could be applied to other ongoing efficacy studies. Further, the analysis of RV144 breakthrough infections was consistent with GES being lower in the vaccinated infected participants compared to placebos and hence protective. These observations, albeit only significant in the group infected with single founder viruses, strengthen the premise of the RV144 GES being a correlate of reduced risk of infection.

We previously showed in NHP challenge studies that the gene signature correlated with an increased magnitude of functional antibody responses (*Ehrenberg et al., 2019*). Although this geneset is associated with ADCP in NHP, the same analyses were previously not possible in the human RV144 study since this immune response was not reported (*Haynes et al., 2012*). ADCP has since been implicated with vaccine protection in a number of NHP challenge studies (*Ackerman et al., 2018*; *Barouch et al., 2015*; *Barouch et al., 2013*; *Barouch et al., 2018*; *Bradley et al., 2017*; *Neidich et al., 2019*). It is reported that ADCP could be involved in most studies that previously showed antibody-dependent correlates of protection against viruses (*Tay et al., 2019b*). To investigate the effect of the gene signature on the magnitude of ADCP, we performed transcriptomics in samples from a human trial (RV306) that employed the same RV144 regimen. At both day 3 and 2 weeks after the 4th vaccine corresponding to the last RV144 vaccine dose, this signature was associated with an increased magnitude of ADCP responses. A strong correlation was also observed between GES from the ADCP enriched genes and the vaccine protection genes in RV144. The ADCP GES at both day 3 and week 2 after the last vaccination was able to predict ADCP magnitude at peak immunity. Similarly, a GES of the 82 overlapping genes between the two time points was also able to predict ADCP magnitude. This finding would suggest that measuring gene expression 3 days after vaccination in preclinical and clinical trials may be used as a tool for titrating effective responses. Although the 82 genes did not have membership in a previously described Fc-gamma receptor signaling pathway, a GO query identified *SIRPA*, *SIRPB1*, *RAB20*, and *TYROBP* in the phagocytosis GO term (*Swanson and Hoppe, 2004*) (http://amigo.geneontology.org/amigo/term/GO:0006909#display-lineage-tab). These four genes were also connected to an additional 44 genes from the total 178 genes enriched in at least one of our GSEA analyses, supporting a possible role in phagocytosis (*Supplementary file 1d*). SIRPA and SIRPB are signal regulatory proteins that are expressed on myeloid cells, and the former is known to bind to CD47 to regulate migration and phagocytosis (*Barclay, 2009*). RAB20 is a member of the Rab GTPase family and is involved in phagosome maturation (*Seto et al., 2011*). Finally, TYROBP

had the maximum connections in a network of these genes, was present in the largest cluster, and was related to other highly connected genes expressed in effector cells such as monocytes or neutrophils. TYROBP (DAP12) is a cytosolic adaptor that associates with triggering receptors expressed on myeloid cells (TREMs) to promote phagocytosis (*N'Diaye et al., 2009*). Genes such as *TNFSF13* and *BHLHE40* that were previously identified as protective against SIV/SHIV acquisition in the NHP model were also present in the same cluster, supporting a similar function (*Ehrenberg et al., 2019*). Taken together, these findings provide evidence supporting the antibody-mediated effector function as a potential mechanistic basis of this protective gene signature.

The enriched genes from ADCP and infection risk in multiple studies of both NHP and human were involved in overlapping functions related to leukocyte activation, lysosomal degradation, and immune stimulation by cytokines. The 63 genes from the RV144 signature had the highest membership in the myeloid leukocyte activation pathway, perhaps alluding to the cellular origin of this signature. The specific genes in the geneset that associated with the greatest odds of reduced risk of infection including *SEMA4A*, *CTSD*, *CD68*, and *GAA* were all members of this pathway, but not *TNFSF13* (*APRIL*) which was the most protective gene in the NHP studies. Although the geneset of interest was first seen in sorted B cells from vaccinated NHP, it was subsequently identified in transcriptomic data from PBMC in the RV144 study (*Ehrenberg et al., 2019*). While samples were exhausted from the RV144 primary data set, the RV306 clinical trial that employed the same ALVAC/protein vaccine regimen gave us a unique chance to explore the cellular origin of the RV144 signature using single-cell transcriptomics. Single-cell surface expression data revealed that the majority of genes were expressed in monocytes, which was not surprising given the fact that this geneset was originally defined as genes downregulated in B cells compared to monocytes after influenza vaccination (*Nakaya et al., 2011*). While our initial study found this signature in sorted B cells from the NHP challenge studies, single-cell data provides further insight that monocytes could be the cellular origin of these genes in the RV144 study. Although monocytes were classified as mononuclear phagocytes almost 50 years ago, assays designed to specifically measure monocyte ADCP were not widely used in the context of vaccination until a few years ago (*van Furth et al., 1972*). While monocytes have been implicated in vaccine-induced protection in preclinical vaccine trials of SIV challenge, our findings in human trials at the single-cell level provide greater impetus to explore the role of other non-lymphoid cell populations on HIV-1 VE (*Gorini et al., 2020*; *Vaccari et al., 2018*). Though we think that monocytes are important in the vaccine responses observed in RV144, it would be remiss not to mention that the effect of granulocytes (including neutrophils) in response to vaccination is missed when transcriptomics is performed in PBMC compared to blood. Regardless of the cellular origin, we think this set of 200 genes with a coordinated expression may not be specific to a cell type, but might mark a certain biological state, such as response to a cytokine, and can be identified even in PBMC and blood samples. Other than the phagocytic cell, antibody and Fc receptor diversity can also influence ADCP mediated immune responses to viral pathogens and are elements that warrant further study and may potentially be manipulated to improve VE (*Chung and Alter, 2017*; *Geraghty et al., 2019*; *Tay et al., 2019b*).

Our data demonstrate the potential to discover novel protective correlates using an approach that mines transcriptomic data in multiple preclinical and clinical trials. Unbiased transcriptome-wide analyses are able to identify biological perturbations that associate with vaccine protection even when differences are small, but credibility can only be strengthened by replicating findings across multiple studies. Gene signatures that associate not only with vaccine protection but with specific immune responses can be a prospective tool to evaluate vaccine effectiveness even prior to challenge or infection. Developing analytical tools that can interface with phenotypes such as vaccine protection across human and preclinical studies can allow for more systematic meta-analyses of data emerging from the ongoing, non-efficacious, or halted HIV vaccine clinical trials (*Gray et al., 2021*; *NIH, 2020*; *NIH, 2021*). We propose that assessment of such gene signatures with immune responses in human immunogenicity trials could provide orthogonal insight for down-selection of vaccine candidates. Identifying overlapping immune correlates could be pivotal to making discoveries that may allow for licensure and subsequent bridging studies of an effective HIV vaccine.

# Materials and methods
## Study design
The aim of the study was post hoc analyses of a protective gene expression signature identified previously in five SIV/HIV vaccine studies with efficacy and immune response data (*Ehrenberg et al., 2019*). To enable interpretation of this gene signature, bulk RNA-seq, scRNA-seq, and functional data were generated in clinical samples from the RV306 and HVTN 505 human trials. The RV306 vaccine trial was conducted in Thailand and all participants received the primary RV144 ALVAC/gp120 vaccine series, with additional late boosts assigned to specific groups (*Pitisuttithum et al., 2020*). Bulk RNA-seq was performed in 24 participants 2 weeks after the RV144 vaccine regimen (week 26). Additionally, RNA-seq was also performed 3 days after the same primary endpoint. The HVTN 505 trial used a DNA/rAd5 vaccine regimen to test safety and efficacy in a US population (*Hammer et al., 2013*). PBMC collected 1 month after the final immunization (month 7) was available from 47 vaccines in the HVTN 505 study for RNA-seq (*Hammer et al., 2013*). The infection status of the vaccinees (22 cases and 25 controls) was categorized based on infection status between months 7 and 24. Microarray transcriptome data from PBMC and immune response data for 170 vaccinated individuals from the RV144 study at time point 2 weeks post last vaccination was used for correlates analyses (*Fourati et al., 2019*; *Haynes et al., 2012*). All studies were approved by the participating local and international institution review boards. Informed consent was obtained from all participants in the different trials included in this study (*Hammer et al., 2013*; *Pitisuttithum et al., 2020*).

## Bulk transcriptomics
RNA was extracted from sorted B cells (Aqua live/dead⁻CD20⁺CD3⁻) and monocytes (Aqua live/dead⁻CD20⁻CD3⁻CD56⁻HLA-DR⁺CD14⁺) from PBMC of HVTN 505 vaccinees using RNAzolRT (MRC Inc) as per recommendations from the manufacturer. For the preparation of mRNA libraries, polyadenylated transcripts were purified on oligo-dT magnetic beads, fragmented, reverse transcribed using random hexamers, and incorporated into barcoded cDNA libraries based on the Illumina TruSeq platform. Next, libraries were validated by electrophoresis, quantified, pooled, and clustered on Illumina TruSeq v2 flow cells. Clustered flow cells were sequenced on an Illumina HiSeq (2000/4000) using 2×75 base paired-end runs. Total RNA from RV306 participants was extracted from whole blood collected in PAXgene Blood RNA tubes using associated RNA extraction (both QIAGEN; Germantown, MD) and GlobinClear purification kits (Thermo Fisher Scientific; Waltham, MA) as per the manufacturer's suggestions. RNA-seq was performed using the SMART-Seq technology (*Picelli et al., 2014*; *Ramsköld et al., 2012*). Briefly, cDNA was generated from 10 ng of RNA using the SMART-Seq v4 UltraLow Input RNA Prep Kit (Takara Bio Inc) as per the manufacturer's suggestions, with control RNA spiked-in (Thermo Fisher Scientific). Sequencing libraries were generated using the Nextera XT DNA Sample Prep Kit (Illumina, San Diego, CA). Concentration of each sample in the pooled libraries was determined using the paired-end 300-cycle MiSeq Reagent Nano Kit v2 (2×150 bp) on a MiSeq instrument (both Illumina). Next-generation sequencing was performed on a final adjusted library pool using the paired-end 300-cycle NovaSeq 6000 S2 XP Reagent Kit (2×150 bp) on a NovaSeq instrument (both Illumina) as per the manufacturer's instructions. Fastp v0.19.7 and Trimmomatic v0.33 with default parameters were used to trim low-quality bases from both ends of each read (*Bolger et al., 2014*; *Chen et al., 2018*). Trimmed reads were aligned to the human genome (GRCh38 build 88–92) using HISAT2 v2.1.0 or the STAR aligner (v2.4.2a) and HTSeq (v0.6.1–0.9.1) was used for counting (*Dobin et al., 2013*; *Kim et al., 2015*; *Anders et al., 2015*). Trimmed mean of M-values normalization method, as implemented in the R package edgeR, was used for normalization (*Robinson et al., 2010*).

## Single-cell transcriptomics
Simultaneous evaluation of mRNA and cell surface expression from single cells was performed using feature barcoding (FB) technology from 10× Genomics, based on the CITE-seq technology (*Stoeckius et al., 2017*). Cell hashing (HTO) was used in conjunction with the 10× Genomics 5′V(D)J Feature Barcoding Kit to generate single-cell mRNA gene expression (GEX) and antibody-derived tag (ADT) libraries (*Stoeckius et al., 2017*; *Stoeckius et al., 2018*). Briefly, PBMC from 12 samples were hashed using TotalSeq-C anti-human Hashtag antibodies and combined into two batches. In each batch, surface proteins were stained with a cocktail of 53 TotalSeq-C antibodies (BioLegend). Antibody concentrations were either predetermined by titration (*Kotliarov et al., 2020*) or used at a default

concentration. 50,000 cells from each batch were loaded onto each of four wells of a Chromium chip, and GEX and ADT (HTO and FB) libraries were constructed following the manufacturer's protocol. Libraries were pooled and quantitated using a MiSeq Nano v2 reagent cartridge. Final libraries were sequenced on the NovaSeq 6000, S4 reagent cartridge (2×100 bp) (Illumina).

## CITE-seq data analyses

FASTQ files were demultiplexed with bcl2fastq v2.20 (Illumina). Alignment and counting were performed using Cell Ranger v3.1.0 (10× Genomics) and the human reference files provided by 10× Genomics (human genome GRCh38 and Ensembl annotation v93). The average number of genes per cell was 1453 and the average number of unique molecular identifiers was 4248. The mean read depth per cell was approximately 65,000–84,000. The minimum fraction of reads mapped to the genome was 88 % and sequencing saturation was above 85 % for all lanes, with an average of 88 %. The computational analysis of ADT data was performed using the Seurat v3.1 package (*Stuart et al., 2019*). HTO expression matrices were CLR (Centered Log-Ratio) normalized and demultiplexed using MULTIseqDemux. The FB matrices from the Seurat objects were split into cell-positive and negative droplet matrices using the HTO demultiplexing results, and were used for DSB (Denoised and Scaled by Background) normalization (*Kotliarov et al., 2020*) (https://cran.r-project.org/web/packages/dsb/index.html). Only cells with <10% mitochondrial genes were retained, and cells were assigned to specific donors using the HTO demultiplexing results. A total of 53,777 single cells remained after the quality control process. The gene expression matrices for all samples were normalized and integrated into a single object in Seurat (*Stuart et al., 2019*). Based on the workflow described in Kotliarov et al., a distance matrix was generated from cell surface protein features (*Kotliarov et al., 2020*). This matrix was used for shared-nearest-neighbor finding and clustering at resolution=0.5. Neighbor finding and clustering were performed on the integrated gene expression data at a resolution=0.75 and dimensions=1:30. A tSNE (t-distributed stochastic neighbor embedding) was generated from the protein data PCA. Seurat was used to generate a heatmap, dotplot, and featureplots. Differential gene expression testing was performed within each cluster between the high and low ADCP groups using Seurat's FindMarkers function. ADCP DEG was filtered to genes with >10% expression in either group, a log fold change >0.25, and a Bonferroni p<0.05.

## ADCP assay

The antibody effector function ADCP was measured as previously described (*Ackerman et al., 2011*; *Tay et al., 2019a*; *Tay et al., 2016*). Briefly, A244 gp120 Env-coated fluorescent beads were incubated at 37 °C for 2 hr with diluted plasma (1:50) collected at week 26, 2 weeks after administration of the RV144 vaccination series. Anti-CD4 monoclonal antibody-treated THP-1 cells (human monocytic cell line; ATCC TIB-202) (treated for 15 min at 4 °C) were added to immune complexes and spinoculated for 1 hr at 4 °C to allow phagocytosis to occur. Supernatant was removed, cells were washed, and fixed in paraformaldehyde. Phagocytosis was measured by flow cytometry and a phagocytosis score was calculated as follows: phagocytosis score=(% pos*MFI of Sample)/(% pos*MFI of no-antibody PBS control). The HIV-1 CD4 binding-site broadly neutralizing antibody (bnAb), CH31, was used as a positive control, and the influenza receptor binding site-specific bnAb, CH65, was used as a negative control. Results are representative of two independent experiments.

## Pathway analyses

Association of the protective gene signature with infection (HVTN 505) or magnitude of median ADCP (RV306) responses were analyzed using the Gene Set Enrichment Analysis (GSEA) method as described previously (*Ehrenberg et al., 2019*; *Subramanian et al., 2005*). GSEA was performed on vaccinated HVTN 505 participants at the visit seven time points, 1 month after the last immunization. RNA-seq was performed on samples prior to infection, but participants were categorized based on their infection status. GSEA was performed on 45 RV306 RNA-seq samples that also had ADCP scores obtained at the week 26 (week 2 after the 4th vaccination) time point. Participants were categorized into high and low ADCP groups based on the median values of ADCP measured in a total of 79 vaccinated participants. The RNA-seq gene expression values at the day 3 and week 2 time points were then analyzed for gene enrichment using a gene set of 200 genes, obtained from the Broad Institute (GSE29618_BCELL_VS_MONOCYTE_DAY7_FLU_VACCINE_DN), between the two groups

of samples. The gene signature of interest was considered significantly enriched using a threshold of NES ≥1.4 and p<0.001 as described previously (*Ehrenberg et al., 2019*). The 178 genes enriched with ADCP or infection status in any of the nine analyses, as well as the 82 genes overlapping in enrichment between the two RV306 time points, were used as search terms in GeneMANIA in the Cytoscape software (*Montojo et al., 2010*; *Warde-Farley et al., 2010*). We selected connections such as co-expression, co-localization, pathway, and physical interactions, as well as Reactome and MSigDB for Attributes. Zero additional genes and up to 10 additional attributes were found with GO biological process-based weighting. The genes in the nine-analysis network were clustered further using the MCODE algorithm in the clusterMaker2 Cytoscape plugin with default settings (*Morris et al., 2011*). Pathway enrichment analysis of the 63 genes enriched in the RV144 infection analysis was performed using Metascape with default parameters, database v20210201 (*Zhou et al., 2019*).

## Correlates of protection

Composite GES was computed as the average of standardized expression of normalized enriched genes in the gene signature in different vaccine studies. The samples in each vaccine study were grouped into outcomes after challenge or infection status after immunization (*Barouch et al., 2015*; *Barouch et al., 2018*; *Rerks-Ngarm et al., 2009*). Logistic regression was used for evaluating the association between GES and HIV-1 infection in the RV144 study. The fitting methods accommodate the two-phase sampling design via maximum likelihood estimation (*Breslow and Holubkov, 1997*). Cumulative HIV-1 incidence curves were plotted for the three subgroups of vaccine recipients defined by tertiles into the lower, middle, and upper third of the GES (Low, Medium, and High subgroups), as well as for the entire placebo group HIV negative at week 24 (N=6267 subjects) for reference. These curves were estimated using the Kaplan-Meier method with inverse probability weighting that accounted for the sampling design. Next, VE for the GES subgroups versus the entire placebo group was estimated as one minus the odds of infection in vaccine recipients with Low/Medium/High response divided by the odds of infection in the entire placebo group HIV-1 negative at week 24 of enrollment in the study. The RV144 prediction analysis was implemented by logistic regression. The data set was randomly split into training and testing sets in a 7:3 ratio, while retaining class distributions within the groups. The training data set consisted of 119 individuals while the test data set consisted of 51 individuals. A logistic regression of GES was fit on to the training data set (*Prentice et al., 2015*). The model's discriminative ability was evaluated by generating a ROC curve and the corresponding AUC on the test data set. The prediction accuracy of the model was also assessed on the test data set. The probability that gives minimum misclassification error was chosen as the cutoff. This process was repeated 1000 times and the distribution of the resulting AUC and accuracy were demonstrated by a histogram with a density curve. Similar analysis was performed using a GES computed from the 32 genes that were significantly associated with HIV acquisition.

Among 121 RV144 participants who became infected during the trial and had their HIV-1 genome sequenced at diagnosis, 43 had GES measurements computed from microarray data (*Fourati et al., 2019*; *Rolland et al., 2012*). Vaccine and placebo groups were compared overall and after stratifying infections with single HIV-1 founders.

## Other statistical analyses

Logistic regression that accounted for the sampling design was used for the univariate analyses of the 63 enriched genes. A radar plot of the significant genes was generated to illustrate OR and 95 % CIs. All ORs were reported per 1-SD increase. Significant genes resulted from univariate logistic regressions of the 63 enriched genes were further analyzed with a multivariate stepwise logistic regression to identify genes that were independently associated with HIV protection. Akaike information criterion was used to identify the optimal set of genes. The expressed enriched genes associated with higher magnitude of ADCP in RV306 at day 3 and 2 weeks post the RV144 vaccine regimen were used to compute the ADCP GES in RV144. Spearman correlation was calculated between the ADCP GES from the two time points and the infection GES, respectively.

For prediction analyses, a GES was computed using the 93 genes enriched at the day 3 time point associating with magnitude of ADCP in RV306. The performance of the classifier was assessed using AUC with 95% CI. This model was then tested at the week 2 time point. The prediction accuracy of the model was also assessed on the week 2 test data set. The probability that gives minimum

misclassification error was chosen as the cutoff. Similar analysis was done using the 82 genes overlapping between the 118 enriched genes from week 2 and the 93 enriched genes from day 3 post the 4th RV144 vaccination series that were associated with ADCP in RV306. The average expression values for 82 overlapping genes in RV306 at the day 3 and week 2 post vaccination time points were stratified by ADCP scores to generate a heatmap with the R package pheatmap. Values for each gene were scaled and the resulting z-scores were hierarchically clustered using the 'complete' method.

All descriptive and inferential statistical analyses were performed using GraphPad Prism 8 (GraphPad Software) and R 3.6.1 (or later) software packages. Comparison of groups was performed using Mann-Whitney tests or t-tests when assumptions were met. All logistic regression models were adjusted for gender and baseline risk behavior and one significant principal component axis (*Haynes et al., 2012*; *Prentice et al., 2015*). A two-sided p-value of less than 0.05 was considered significant. The Benjamini and Hochberg method was used to calculate false discovery rate-adjusted p-values for multiple testing corrections.

## Acknowledgements

The authors would like to thank the volunteers and staff of the RV306, RV144, and HVTN 505 clinical trials. The authors also acknowledge DeAnna Tenney and Derrick Goodman, Duke University for expert technical assistance. The views expressed are those of the authors and should not be construed to represent the positions of the US Army or the US Department of Defense (DOD). This work was supported by a cooperative agreement (W81XWH-07-2-0067) between the Henry M Jackson Foundation for the Advancement of Military Medicine, Inc, and the DOD. This research was funded in part by the US National Institute of Allergy and Infectious Disease.

## Additional information

### Funding

| Funder | Grant reference number | Author |
|---|---|---|
| Henry M. Jackson Foundation | W81XWH-07-2-0067 | Shida Shangguan<br>Philip K Ehrenberg<br>Aviva Geretz<br>Lauren Yum<br>Gautam Kundu<br>Kelly May<br>Eric Lewitus<br>Morgane Rolland<br>Nelson L Michael<br>Sandhya Vasan<br>Rasmi Thomas |
| National Institute of Allergy and Infectious Diseases | | Shida Shangguan<br>Philip K Ehrenberg<br>Aviva Geretz<br>Lauren Yum<br>Gautam Kundu<br>Eric Lewitus<br>Morgane Rolland<br>Nelson L Michael<br>Sandhya Vasan<br>Rasmi Thomas |

The funders had no role in study design, data collection and interpretation, or the decision to submit the work for publication.

### Author contributions

Shida Shangguan, Formal analysis, Methodology, Visualization, Writing - original draft, Writing - review and editing; Philip K Ehrenberg, Lauren Yum, LaTonya D Williams, Methodology, Writing - original draft, Writing - review and editing; Aviva Geretz, Methodology, Software, Visualization, Writing - original draft, Writing - review and editing; Gautam Kundu, Methodology, Software, Visualization, Writing - review and editing; Kelly May, Krystelle Nganou-Makamdop, Methodology; Slim

Fourati, Investigation, Software; Sheetal Sawant, Writing - review and editing; Eric Lewitus, Investigation, Writing - original draft; Punnee Pitisuttithum, Sorachai Nitayaphan, Suwat Chariyalertsak, Supachai Rerks-Ngarm, Resources; Morgane Rolland, Investigation, Writing - review and editing; Daniel C Douek, Georgia D Tomaras, Investigation, Resources; Peter Gilbert, Investigation; Nelson L Michael, Funding acquisition, Resources; Sandhya Vasan, Funding acquisition, Resources, Writing - review and editing; Rasmi Thomas, Conceptualization, Formal analysis, Investigation, Methodology, Project administration, Supervision, Visualization, Writing - original draft, Writing - review and editing

## Author ORCIDs
Philip K Ehrenberg http://orcid.org/0000-0002-8695-4301
Slim Fourati http://orcid.org/0000-0001-6609-7587
Rasmi Thomas http://orcid.org/0000-0002-2116-2418

## Decision letter and Author response
Decision letter https://doi.org/10.7554/eLife.69577.sa1
Author response https://doi.org/10.7554/eLife.69577.sa2

# Additional files

## Supplementary files
• Supplementary file 1. Additional tables supporting the findings in this study. (a) Number of enriched genes from the geneset in different comparisons from multiple studies. (b) Number of enriched genes from the geneset in different comparisons from multiple studies. (c) Relationship between genes associating with ADCP in this study and previously identified pathways. (d) Enriched ADCP genes that belong to the gene ontology phagocytosis pathway.

• Transparent reporting form

• Source code 1. Code to generate figures from "Monocyte-derived transcriptome signature indicates antibody-dependent cellular phagocytosis as a potential mechanism of vaccine-induced protection against HIV-1".

## Data availability
All code and data generated or analyzed during this study are included with the manuscript and supporting files. Source code to generate the figures can be accessed at https://github.com/thomaslab-MHRP/eLife_2021. Source data files have been provided for all data used in this study, including CITE-seq and gene expression matrix for all studies are available at figshare 10.6084/m9.figshare.c.5416197. The RNA-seq gene expression data for RV306 and HVTN 505 studies are available in the National Center for Biotechnology Information Gene Expression Omnibus (GEO) under accession numbers: "GSE181932" and "GS1E181859" respectively.

The following data set was generated:

| Author(s) | Year | Dataset title | Dataset URL | Database and Identifier |
|---|---|---|---|---|
| Geretz A, Shangguan S, Thomas R | 2021 | Monocyte-derived transcriptome signature indicates antibody-dependent cellular phagocytosis as a potential mechanism of vaccine-induced protection against HIV-1 | https://doi.org/10.6084/m9.figshare.c.5416197 | figshare, 10.6084/m9.figshare.c.5416197 |
| Hu J, Douek DC, Nganou-Makamdop K, Fourati S | 2021 | Transcriptional profiling of HIV-negative participants before and after vaccination with an DNA/rAd5 vaccine | https://www.ncbi.nlm.nih.gov/geo/query/acc.cgi?acc=GSE181859 | NCBI Gene Expression Omnibus, GSE181859 |

*Continued on next page*

*Continued*

| Author(s) | Year | Dataset title | Dataset URL | Database and Identifier |
|---|---|---|---|---|
| Shangguan S, Geretz A, Thomas R | 2021 | Transcriptomic profiling of human peripheral blood samples collected three days or two weeks after the primary RV144 ALVAC/gp120 vaccine series | https://www.ncbi.nlm.nih.gov/geo/query/acc.cgi?acc=GSE181932 | NCBI Gene Expression Omnibus, GSE181932 |

The following previously published data sets were used:

| Author(s) | Year | Dataset title | Dataset URL | Database and Identifier |
|---|---|---|---|---|
| Ehrenberg PK, Shangguan S, Issac B, Alter G, Geretz A, Izumi T, Bryant C, Eller MA, Wegmann F, Apps R, Creegan M, Bolton DL, Sekaly RP, Robb ML, Gramzinski RA, Pau MG, Schuitemaker H, Barouch DH, Michael NL, Thomas R | 2019 | Vaccine-induced gene expression signature correlates with protection against SIV and HIV in multiple trials | https://www.ebi.ac.uk/ena/browser/view/PRJEB31297 | EBI, PRJEB31297 |
| Fourati S, Ribeiro SP, Blasco Tavares Pereira Lopes F, Talla A, Lefebvre F, Cameron M, Kaewkungwal J, Pitisuttithum P, Nitayaphan S, Rerks-Ngarm S, Kim JH, Thomas R, Gilbert PB, Tomaras GD, Koup RA, Michael NL, McElrath MJ, Gottardo R, Sékaly R-P | 2019 | Transcriptomic profiling of RV144 trial participants | https://www.ncbi.nlm.nih.gov/geo/query/acc.cgi?acc=GSE103740 | NCBI Gene Expression Omnibus, GSE103740 |

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

# Appendix 1

**Appendix 1**-key resources table

| Reagent type (species) or resource | Designation | Source or reference | Identifiers | Additional information |
|---|---|---|---|---|
| Antibody | Anti-Human CD1c (Mouse monoclonal) | BioLegend | Cat# 331547, Clone L161, RRID:AB_2800871 | 10× Genomics FB Ab pool: 0.25 µg (1:180) |
| Antibody | Anti-Human CD163 (Mouse monoclonal) | BioLegend | Cat# 333637, Clone GHI/61, RRID:AB_2810510 | 10× Genomics FB Ab pool: 1 µg (1:45) |
| Antibody | Anti-Human CD141 (Mouse monoclonal) | BioLegend | Cat# 344125, Clone M80, RRID:AB_2810541 | 10× Genomics FB Ab pool: 0.5 µg (1:90) |
| Antibody | Anti-Human CD11a (Mouse monoclonal) | BioLegend | Cat# 350617, Clone TS2/4, RRID:AB_2800935 | 10× Genomics FB Ab pool: 1 µg (1:45) |
| Antibody | Anti-Human CD197 (Mouse monoclonal) | BioLegend | Cat# 353251, Clone G043H7, RRID:AB_2800943 | 10× Genomics FB Ab pool: 1 µg (1:45) |
| Antibody | Anti-Human CD14 (Mouse monoclonal) | BioLegend | Cat# 301859, Clone M5E2, RRID:AB_2800736 | 10× Genomics FB Ab pool: 1 µg (1:45) |
| Antibody | Anti-Human CD16 (Mouse monoclonal) | BioLegend | Cat# 302065, Clone 3G8, RRID:AB_2800738 | 10× Genomics FB Ab pool: 1 µg (1:45) |
| Antibody | Anti-Human CD19 (Mouse monoclonal) | BioLegend | Cat# 302265, Clone HIB19, RRID:AB_2800741 | 10× Genomics FB Ab pool: 0.5 µg (1:90) |
| Antibody | Anti-Human CD45RO (Mouse monoclonal) | BioLegend | Cat# 304259, Clone UCHL1, RRID:AB_2800766 | 10× Genomics FB Ab pool: 1 µg (1:45) |
| Antibody | Anti-Human CD2 (Mouse monoclonal) | BioLegend | Cat# 309231, Clone TS1/8, RRID:AB_2810464 | 10× Genomics FB Ab pool: 0.125 µg (1:360) |
| Antibody | Anti-Human CD138 (Mouse monoclonal) | BioLegend | Cat# 356539, Clone MI15, RRID:AB_2810567 | 10× Genomics FB Ab pool: 1 µg (1:45) |
| Antibody | Anti-Human CD303 (Mouse monoclonal) | BioLegend | Cat# 354241, Clone 201 A, RRID:AB_2814295 | 10× Genomics FB Ab pool: 1 µg (1:45) |
| Antibody | Anti-Human CD56 (Mouse monoclonal) | BioLegend | Cat# 362559, Clone 5.1h11, RRID:AB_2801002 | 10× Genomics FB Ab pool: 1 µg (1:45) |
| Antibody | Anti-Human CD4 (Mouse monoclonal) | BioLegend | Cat# 300567, Clone RPA-T4, RRID:AB_2800725 | 10× Genomics FB Ab pool: 1 µg (1:45) |
| Antibody | Anti-Human CD3 (Mouse monoclonal) | BioLegend | Cat# 300479, Clone UCHT1, RRID:AB_2800723 | 10× Genomics FB Ab pool: 1 µg (1:45) |
| Antibody | Anti-Human CD45RA (Mouse monoclonal) | BioLegend | Cat# 304163, Clone HI100, RRID:AB_2800764 | 10× Genomics FB Ab pool: 1 µg (1:45) |
| Antibody | Anti-Human CD39 (Mouse monoclonal) | BioLegend | Cat# 328237, Clone A1, RRID:AB_2800853 | 10× Genomics FB Ab pool: 1 µg (1:45) |
| Antibody | Anti-Human CD279 (Mouse monoclonal) | BioLegend | Cat# 329963, Clone EH12.2H7, RRID:AB_2800862 | 10× Genomics FB Ab pool: 1 µg (1:45) |
| Antibody | Anti-Human CD8 (Mouse monoclonal) | BioLegend | Cat# 344753, Clone SK1, RRID:AB_2800922 | 10× Genomics FB Ab pool: 1 µg (1:45) |
| Antibody | Anti-Human CD27 (Mouse monoclonal) | BioLegend | Cat# 302853, Clone O323, RRID:AB_2800747 | 10× Genomics FB Ab pool: 0.25 µg (1:180) |
| Antibody | Anti-Human CD20 (Mouse monoclonal) | BioLegend | Cat# 302363, Clone 2H7, RRID:AB_2800743 | 10× Genomics FB Ab pool: 1 µg (1:45) |
| Antibody | Anti-Human HLA-A/B/C (Mouse monoclonal) | BioLegend | Cat# 311449, Clone W6/32, RRID:AB_2800816 | 10× Genomics FB Ab pool: 1 µg (1:45) |
| Antibody | Anti-Human IgM (Mouse monoclonal) | BioLegend | Cat# 314547, Clone MHM-88, RRID:AB_2800835 | 10× Genomics FB Ab pool: 0.5 µg (1:90) |
| Antibody | Anti-Human CD127 (Mouse monoclonal) | BioLegend | Cat# 351356, Clone A019D5, RRID:AB_2800937 | 10× Genomics FB Ab pool: 0.5 µg (1:90) |
| Antibody | Anti-Human CD195 (Rat monoclonal) | BioLegend | Cat# 359137, Clone J418F1, RRID:AB_2810570 | 10× Genomics FB Ab pool: 0.25 µg (1:180) |

*Continued on next page*

*Continued*

**Appendix 1**-key resources table

| Reagent type (species) or resource | Designation | Source or reference | Identifiers | Additional information |
|---|---|---|---|---|
| Antibody | Anti-Human HLA-DR (Mouse monoclonal) | BioLegend | Cat# 307663, Clone L243, RRID:AB_2800795 | 10× Genomics FB Ab pool: 1 µg (1:45) |
| Antibody | Anti-Human IgG (Fc) (Rat monoclonal) | BioLegend | Cat# 410727, Clone M1310G05, RRID:AB_2801087 | 10× Genomics FB Ab pool: 1 µg (1:45) |
| Antibody | Anti-Human TCR Vd2 (Mouse monoclonal) | BioLegend | Cat# 331435, Clone B6, RRID:AB_2800864 | 10× Genomics FB Ab pool: 1 µg (1:45) |
| Antibody | Anti-Human TCR Va7.2 (Mouse monoclonal) | BioLegend | Cat# 351735, Clone 3C10, RRID:AB_2810556 | 10× Genomics FB Ab pool: 1 µg (1:45) |
| Antibody | Anti-Human TCR Va24-Ja18 (Mouse monoclonal) | BioLegend | Cat# 342925, Clone 6B11, RRID:AB_2810539 | 10× Genomics FB Ab pool: 1 µg (1:45) |
| Antibody | Anti-Human TCR g/d (Mouse monoclonal) | BioLegend | Cat# 331231, Clone B1, RRID:AB_2814199 | 10× Genomics FB Ab pool: 1 µg (1:45) |
| Antibody | Anti-Human TCR Vg9 (Mouse monoclonal) | BioLegend | Cat# 331313, Clone B3, RRID:AB_2814203 | 10× Genomics FB Ab pool: 1 µg (1:45) |
| Antibody | Anti-Human CD7 (Mouse monoclonal) | BioLegend | Cat# 343127, Clone CD7-6B7, RRID:AB_2800914 | 10× Genomics FB Ab pool: 1 µg (1:45) |
| Antibody | Anti-Human CD11c (Mouse monoclonal) | BioLegend | Cat# 371521, Clone S-HCL-3, RRID:AB_2801018 | 10× Genomics FB Ab pool: 0.125 µg (1:360) |
| Antibody | Anti-Human CD185 (Mouse monoclonal) | BioLegend | Cat# 356939, Clone J252D4, RRID:AB_2800968 | 10× Genomics FB Ab pool: 1 µg (1:45) |
| Antibody | Anti-Human CD1d (Mouse monoclonal) | BioLegend | Cat# 350319, Clone 51.1, RRID:AB_2800934 | 10× Genomics FB Ab pool: 1 µg (1:45) |
| Antibody | Anti-Human IgD (Mouse monoclonal) | BioLegend | Cat# 348245, Clone IA6-2, RRID:AB_2810553 | 10× Genomics FB Ab pool: 0.5 µg (1:90) |
| Antibody | Anti-Human CD11b (Mouse monoclonal) | BioLegend | Cat# 301359, Clone ICRF44, RRID:AB_2800732 | 10× Genomics FB Ab pool: 1 µg (1:45) |
| Antibody | Anti-Human CD62L (Mouse monoclonal) | BioLegend | Cat# 304851, Clone DREG-56, RRID:AB_2800770 | 10× Genomics FB Ab pool: 0.125 µg (1:360) |
| Antibody | Anti-Human CD66a/c/e (Mouse monoclonal) | BioLegend | Cat# 342325, Clone ASL-32, RRID:AB_2810538 | 10× Genomics FB Ab pool: 1 µg (1:45) |
| Antibody | Anti-Human CD15 (Mouse monoclonal) | BioLegend | Cat# 323053, Clone W6D3, RRID:AB_2800847 | 10× Genomics FB Ab pool: 1 µg (1:45) |
| Antibody | Anti-Human CD32 (Mouse monoclonal) | BioLegend | Cat# 303225, Clone FUN-2, RRID:AB_2814129 | 10× Genomics FB Ab pool: 0.5 µg (1:90) |
| Antibody | Anti-Human CD57 (Mouse monoclonal) | BioLegend | Cat# 393321, Clone QA17A04, RRID:AB_2801030 | 10× Genomics FB Ab pool: 1 µg (1:45) |
| Antibody | Anti-Human CD73 (Mouse monoclonal) | BioLegend | Cat# 344031, Clone AD2, RRID:AB_2800916 | 10× Genomics FB Ab pool: 1 µg (1:45) |
| Antibody | Anti-Human CD123 (Mouse monoclonal) | BioLegend | Cat# 306045, Clone 6H6, RRID:AB_2800789 | 10× Genomics FB Ab pool: 1 µg (1:45) |
| Antibody | Anti-Human Mouse IgG1, k Isotype Ctrl (Mouse monoclonal) | BioLegend | Cat# 400187, Clone MOPC-21, RRID:AB_2888921 | 10× Genomics FB Ab pool: 1 µg (1:45) |
| Antibody | Anti-Human Mouse IgG2a, k Isotype Ctrl (Mouse monoclonal) | BioLegend | Cat# 400293, Clone MOPC-173, RRID:AB_2888922 | 10× Genomics FB Ab pool: 1 µg (1:45) |
| Antibody | Anti-Human Mouse IgG2b, k Isotype Ctrl (Mouse monoclonal) | BioLegend | Cat# 400381, Clone MPC-11, RRID:AB_2888923 | 10× Genomics FB Ab pool: 1 µg (1:45) |
| Antibody | Anti-Human Rat IgG2b, k Isotype Ctrl (Rat monoclonal) | BioLegend | Cat# 400677, Clone RTK4530 | 10× Genomics FB Ab pool: 1 µg (1:45) |
| Antibody | Anti-Human CD28 (Mouse monoclonal) | BioLegend | Cat# 302963, Clone CD28.2, RRID:AB_2800751 | 10× Genomics FB Ab pool: 1 µg (1:45) |
| Antibody | Anti-Human CD161 (Mouse monoclonal) | BioLegend | Cat# 339947, Clone HP-3G10, RRID:AB_2810532 | 10× Genomics FB Ab pool: 1 µg (1:45) |

*Continued on next page*

*Continued*

**Appendix 1**-key resources table

| Reagent type (species) or resource | Designation | Source or reference | Identifiers | Additional information |
|---|---|---|---|---|
| Antibody | Anti-Human CD95 (Mouse monoclonal) | BioLegend | Cat# 305651, Clone DX2, RRID:AB_2800787 | 10× Genomics FB Ab pool: 1 µg (1:45) |
| Antibody | Anti-Human CD38 (Mouse monoclonal) | BioLegend | Cat# 303543, Clone HIT2, RRID:AB_2800758 | 10× Genomics FB Ab pool: 1 µg (1:45) |
| Antibody | Hash1: anti-Human CD298 & b2-microglobulin (Mouse monoclonals) | BioLegend | Cat# 394661, Clone LNH-94; 2M2 RRID:AB_2801031 | 10× Genomics Hash Ab input: 1 µg (1:45) |
| Antibody | Hash2: anti-Human CD298 & b2-microglobulin (Mouse monoclonals) | BioLegend | Cat# 394663, Clone LNH-94; 2M2 RRID:AB_2801032 | 10× Genomics Hash Ab input: 1 µg (1:45) |
| Antibody | Hash3: anti-Human CD298 & b2-microglobulin (Mouse monoclonals) | BioLegend | Cat# 394665, Clone LNH-94; 2M2 RRID:AB_2801033 | 10× Genomics Hash Ab input: 1 µg (1:45) |
| Antibody | Hash4: anti-Human CD298 & b2-microglobulin (Mouse monoclonals) | BioLegend | Cat# 394667, Clone LNH-94; 2M2 RRID:AB_2801034 | 10× Genomics Hash Ab input: 1 µg (1:45) |
| Antibody | Hash7: anti-Human CD298 & b2-microglobulin (Mouse monoclonals) | BioLegend | Cat# 394673, Clone LNH-94; 2M2 RRID:AB_2820043 | 10× Genomics Hash Ab input: 1 µg (1:45) |
| Antibody | Hash8: anti-Human CD298 & b2-microglobulin (Mouse monoclonals) | BioLegend | Cat# 394675, Clone LNH-94; 2M2 RRID:AB_2820044 | 10× Genomics Hash Ab input: 1 µg (1:45) |
| Antibody | Hash9: anti-Human CD298 & b2-microglobulin (Mouse monoclonals) | BioLegend | Cat# 394677, Clone LNH-94; 2M2 RRID:AB_2820045 | 10× Genomics Hash Ab input: 1 µg (1:45) |
| Antibody | Hash10: anti-Human CD298 & b2-microglobulin (Mouse monoclonals) | BioLegend | Cat# 394679, Clone LNH-94; 2M2 RRID:AB_2820046 | 10× Genomics Hash Ab input: 1 µg (1:45) |
| Antibody | FITC anti-Human CD56 (Mouse monoclonal) | BD Biosciences | Cat# 340410, RRID:AB_400025 | FACS (1:25) |
| Antibody | PE Anti-Human CD14 (Mouse monoclonal) | BD Biosciences | Cat# 555398, RRID:AB_395799 | FACS (1:200) |
| Antibody | APC-Cy7 Anti-Human CD3 (Mouse monoclonal) | BD Biosciences | Cat# 557832, RRID:AB_396890 | FACS (1:50) |
| Antibody | Brilliant Violet 570 anti-human CD20 (Monoclonal) | BioLegend | Cat# 302332, RRID:AB_2563805 | FACS (1:50) |
| Antibody | PE-Cyanine5.5 Anti-Human HLA-DR (Mouse monoclonal) | Invitrogen | Cat# MHLDR18, RRID:AB_1500218 | FACS (1:100) |
| Commercial assay or kit | LIVE/DEAD Fixable Aqua Dead Cell Stain Kit | Thermo Fisher Scientific | L34957 | FACS |
| Antibody | Anti-CD4 (Human, monoclonal) | BioLegend | Cat# 344,602 | ADCP Assay: (20 µl/ml) |
| Antibody | CH31 (Human monoclonal) | PMID: 22301150 | | ADCP Assay: Duke Human Vaccine Institute (DHVI) Protein Production Facility (PPF); (50 µg/ml) |
| Antibody | CH65 (Human monoclonal) | PMID: 21825125 | | ADCP Assay: Duke Human Vaccine Institute (DHVI) Protein Production Facility (PPF); (50 µg/ml) |
| Cell line (*Homo sapiens*) | THP-1 | ATCC | Cat# TIB-202 | Identity has been authenticated by STR profiling and mycoplasma contamination was not detected. It is not included in the list of commonly misidentified cell lines. |

