## [Decision Letter]

**Acceptance summary:**

Attempts to produce effective vaccines against HIV have not yet been successful with a lack of understanding of the correlates of protection being a significant limitation. This paper analyses gene expression in a number of human and non-human primate vaccine trials and identifies a profile that appears to correlate with protection from infection. This profile is linked primarily to monocytes and the ability of these cells to mediate antibody dependent cellular phagocytosis. The work has implications for ongoing attempts to generate effective vaccines against HIV and perhaps other viral diseases.

**Decision letter after peer review:**

Thank you for submitting your article "Monocyte-derived transcriptome signature indicates antibody-dependent cellular 1 phagocytosis as the primary mechanism of vaccine-induced protection against HIV-1" for consideration by *eLife*. Your article has been reviewed by 3 peer reviewers, and the evaluation has been overseen by a Reviewing Editor and Päivi Ojala as the Senior Editor. The following individual involved in review of your submission has agreed to reveal their identity: Genoveffa Franchini (Reviewer #2).

Essential revisions:

1) The NHP data on ALVAC/gp120 /MF59 (that was not effective) (Vaccari et al.,2016), as requested by reviewer 2 should be included.

2) Further information should be provided on the protective gene set in the various trials and the overlapping genes in these gene sets. Van Diagrams and Tables summarizing the common or private genes in the different NHP, and human trials would be very informative, as requested by reviewer 2.

3) The title should be tempered, as suggested by two of the reviewers.

4) You should respond to all of the points raised by the reviewers, either by the inclusion of additional data, modifications to the text or in a rebuttal letter.

*Reviewer #1 (Recommendations for the authors):*

In Vaccari et al. Nat Med. 2016 Jul; 22(7): 762-770 the RAS pathway is reported associated with resistance to infection and favorable innate immunity. Is this pathway observed as significant in the trials reported here? How are the pathways reveled here used to interpreted biologically how they can provide better protection based on the function they are involved with?

All human samples described as RV306 are samples are derived 2 weeks after last RV144 vaccination, therefore the trial at that point was still RV144, no reason to call it RV306 and this trial should be left out, as that implies sampling after the additional boosting of RV144 (which, I may be wrong, but it does not seem the case for any of the samples here) and becomes confusing for the reader.

When looking at Methods, it is confusing what is new to this paper and what was already part of previous publications (for instance Ehrenberg et al., 2019). For instance, RNA transcriptomics appear to have been done for the previous publication. If so, it should not be here. The same is true for ADCP that was also reported in Ehrenberg et al., 2019. Methods listed here should only be those used for the new analyses reported here, not those used to generate data already reported in previous publications.

*Reviewer #2 (Recommendations for the authors):*

This is an interesting paper and i provide my suggestions to improve it. Most of all I believe that the choice of the vaccine modalities to be compared needs to be more comprehensive and the data are available.

1)The NHP data on ALVAC/gp120 /alum ((that was efficacious) was analyzed, but not that of the ALVAC/gp120 /MF59 (that was not effective) (Vaccari et al.,2016).

Question to be addressed: Was the "protective" gene set not enriched in the MF59 arm?

This would be particularly important to strengthen the hypothesis put forward by the authors, in light of the failure of human HIV vaccine trial (HVTN702) that used the same vaccine modalities in RV144 except for the adjuvant (MF59 rather than Alum).

2) Germane to the "protective" gene set described here and the conclusion that monocytes are important in protection is a study that tested, in parallel to the DNA prime /ALVAC/gp120/alum boost vaccine (Table 1) to the Ad26 prime / ALVAC /gp120/alum boost. The DNA prime was protective and vaccine efficacy correlated with classical monocyte frequency as well as with a monocyte gene signature. The Ad26 prime was not protective (Vaccari et al., 2018)

Question to be addressed: Was the "protective" gene set not enriched in the Ad26/ALVAC /gp120/alum arm?

3) In the text it is difficult to grasp what is the protective gene set in the various trials and what are the overlapping genes in these gene sets. Van Diagrams and Tables summarizing the common or private geneses in the different NHP, and human trials would be very informative.

4) There are several trials described here and all need to be summarized in Table 1

Table 1 should include also RV306, and the origin of the samples analyzed (whole blood, cultured stimulated and unstimulated cells, PBMC for each study)

5) The title of the paper refers to …' phagocytosis as the primary mechanism of vaccine induced protection'

a) The data on phagocytosis before and after vaccination in this subset of patients should be shown

b) The Title should also be tempered

Monocyte-derived transcriptome signature indicates antibody-dependent cellular phagocytosis…. Please clarify in the text which gene set defines specifically phagocytosis?

…as the primary mechanism of vaccine -induced protection …

The conclusion is based on a correlation of a correlation between two independent trials.

It should not be referred as the primary mechanism…

---

## [Author Response]

Essential revisions:1) The NHP data on ALVAC/gp120 /MF59 (that was not effective) (Vaccari et al.,2016), as requested by reviewer 2 should be included.

The premise of our findings is that in studies where there is partial efficacy, we find a common protective gene signature when comparing between uninfected and infected vaccine recipients. A similar analysis is not possible in the ALVAC-SIV/gp120/MF59 arm mentioned by the reviewer, since all 27 vaccinated animals became infected (Vaccari et al., 2016). We now clarify this at the beginning of the Results section and also by adding a sentence observing that we test this signature across studies/trials that show partial efficacy, and find the signature associates with trials that showed significant protection, but cannot test non-efficacious regimens where all animals/recipients were infected (Discussion section).

2) Further information should be provided on the protective gene set in the various trials and the overlapping genes in these gene sets. Van Diagrams and Tables summarizing the common or private genes in the different NHP, and human trials would be very informative, as requested by reviewer 2.

The overlapping genes in these gene sets are now detailed in an added Supplementary File 1b. This table shows the 200 genes in the SE29618_BCELL_VS_MONOCYTE_DAY7_FLU_VACCINE_DN geneset and the genes that were expressed or enriched in each of the different NHP or human trials. Venn diagrams of overlapping genes, as well as pathway analyses of enriched genes in different studies are shown in new Figures 4A, C-D and 5A.

3) The title should be tempered, as suggested by two of the reviewers.

We agree and have updated the title to “Monocyte-derived transcriptome signature indicates antibody-dependent cellular phagocytosis as a potential mechanism of vaccine-induced protection against HIV-1”.

4) You should respond to all of the points raised by the reviewers, either by the inclusion of additional data, modifications to the text or in a rebuttal letter.

We have responded to all the points raised by the reviewers.

Reviewer #1 (Recommendations for the authors):In Vaccari et al. Nat Med. 2016 Jul; 22(7): 762-770 the RAS pathway is reported associated with resistance to infection and favorable innate immunity. Is this pathway observed as significant in the trials reported here? How are the pathways reveled here used to interpreted biologically how they can provide better protection based on the function they are involved with?

There are several pathways in current literature that associate with innate immunity and favorable responses in specific vaccine studies. We did not choose to look for specific pathway(s) but used an unbiased method to identify the strongest correlates of vaccine efficacy in multiple NHP and human studies. We also investigated immune responses associating with this signature, which showed that ADCP might be a potential lead for mechanistic investigation. The associated pathways show leukocyte activation, lysosomes and signaling by cytokines to be important in these partially protective vaccine responses. We think that such omics approaches can potentially be used to assist with down-selection of promising vaccine candidates in preclinical and early clinical trials, and to identify potential mechanisms of protection that might be targeted in future vaccine designs.

All human samples described as RV306 are samples are derived 2 weeks after last RV144 vaccination, therefore, the trial at that point was still RV144, no reason to call it RV306 and this trial should be left out, as that implies sampling after the additional boosting of RV144 (which, I may be wrong, but it does not seem the case for any of the samples here) and becomes confusing for the reader.

The RV306 trial is independent from the RV144 trial, and there are no overlapping participants. The RV306 trial conducted in Thailand used the same RV144 vaccine series but included additional boosts after the fourth vaccine (Pitisuttithum et al., 2020). We focused on timepoints that overlapped with the original RV144 vaccine series for generating RNA-seq and ADCP data. We now include supplementary file 1a to clarify the different studies described in this manuscript.

When looking at Methods, it is confusing what is new to this paper and what was already part of previous publications (for instance Ehrenberg et al., 2019). For instance, RNA transcriptomics appear to have been done for the previous publication. If so, it should not be here. The same is true for ADCP that was also reported in Ehrenberg et al., 2019. Methods listed here should only be those used for the new analyses reported here, not those used to generate data already reported in previous publications.

We have included in the methods specifically bulk transcriptomics for RNA-seq in the HVTN 505 and RV306 studies, and ADCP assay data for RV306. These are new datasets and have not been published previously, but to clarify further we have included supplementary file 1a showing the new transcriptomic datasets that were generated for this manuscript.

Reviewer #2 (Recommendations for the authors):This is an interesting paper and i provide my suggestions to improve it. Most of all I believe that the choice of the vaccine modalities to be compared needs to be more comprehensive and the data are available.1)The NHP data on ALVAC/gp120 /alum ((that was efficacious) was analyzed, but not that of the ALVAC/gp120 /MF59 (that was not effective) (Vaccari et al.,2016).Question to be addressed: Was the "protective" gene set not enriched in the MF59 arm?

Please see our response above, ‘essential revisions point 1’.

This would be particularly important to strengthen the hypothesis put forward by the authors, in light of the failure of human HIV vaccine trial (HVTN702) that used the same vaccine modalities in RV144 except for the adjuvant (MF59 rather than Alum).2) Germane to the "protective" gene set described here and the conclusion that monocytes are important in protection is a study that tested, in parallel to the DNA prime /ALVAC/gp120/alum boost vaccine (Table 1) to the Ad26 prime / ALVAC /gp120/alum boost. The DNA prime was protective and vaccine efficacy correlated with classical monocyte frequency as well as with a monocyte gene signature. The Ad26 prime was not protective (Vaccari et al., 2018)Question to be addressed: Was the "protective" gene set not enriched in the Ad26/ALVAC /gp120/alum arm?

Please see our response above, Essential revisions point 1. We were unable to test the presence of the gene signature in the Ad26/ALVAC arm from the Vaccari et al. 2018 study, since all 22 monkeys became infected.

3) In the text it is difficult to grasp what is the protective gene set in the various trials and what are the overlapping genes in these gene sets. Van Diagrams and Tables summarizing the common or private geneses in the different NHP, and human trials would be very informative.

We have now included the overlapping genes in an added supplementary file 1b, which shows the 200 genes in the GSE29618_BCELL_VS_MONOCYTE_DAY7_FLU_VACCINE_DN geneset. Status of each gene (enriched, not enriched, or not expressed) is shown for each study. We have additionally created a column in the list which provides the total number of genes expressed and overlapping in each study.

4) There are several trials described here and all need to be summarized in Table 1Table 1 should include also RV306, and the origin of the samples analyzed (whole blood, cultured stimulated and unstimulated cells, PBMC for each study)

We now include a table to summarize all studies specifically investigated in this paper, along with their source of origin, in an added supplementary file 1a. To help differentiate this from studies with vaccine efficacy or protection data, we have separated this from Table 1 that includes the vaccine preclinical and clinical trials.

5) The title of the paper refers to …' phagocytosis as the primary mechanism of vaccine induced protection'a) The data on phagocytosis before and after vaccination in this subset of patients should be shown

We examined the presence of the gene signature with magnitude of ADCP scores only in the post vaccination samples. This analysis is not possible prior to vaccination, since there are only 4 participants (active arm) who met the threshold of positive ADCP scores in the absence of vaccination.

b) The Title should also be temperedMonocyte-derived transcriptome signature indicates antibody-dependent cellular phagocytosis…. Please clarify in the text which gene set defines specifically phagocytosis?…as the primary mechanism of vaccine -induced protection …The conclusion is based on a correlation of a correlation between two independent trials.It should not be referred as the primary mechanism…

We have updated the title per the reviewer’s suggestion (please see above).